# Ditch the Denoiser: Emergence of Noise Robustness in Self-Supervised Learning from Data Curriculum

**Wenquan Lu**[1]    **Jiaqi Zhang**[1]    **Hugues Van Assel**[2]    **Randall Balestriero**[1]

[1]Brown University    [2]Genentech
{wenquan_lu, jiaqi_zhang6, randall_balestriero}@brown.edu
van_assel.hugues@gene.com

## Abstract

Self-Supervised Learning (SSL) has become a powerful solution to extract rich representations from unlabeled data. Yet, SSL research is mostly focused on clean, curated and high-quality datasets. As a result, applying SSL on noisy data remains a challenge, despite being crucial to applications such as astrophysics, medical imaging, geophysics or finance. In this work, we present a fully self-supervised framework that enables noise-robust representation learning without requiring a denoiser at inference or downstream fine-tuning. Our method first trains an SSL denoiser on noisy data, then uses it to construct a denoised-to-noisy data curriculum (i.e., training first on denoised, then noisy samples) for pretraining a SSL backbone (e.g., DINOv2), combined with a teacher-guided regularization that anchors noisy embeddings to their denoised counterparts. This process encourages the model to internalize noise robustness. Notably, the denoiser can be discarded after pretraining, simplifying deployment. On ImageNet-1k with ViT-B under extreme Gaussian noise ($\sigma = 255$, SNR = 0.72 dB), our method improves linear probing accuracy by 4.8% over DINOv2, demonstrating that denoiser-free robustness can emerge from noise-aware pretraining. The code is available at `https://github.com/wenquanlu/noisy_dinov2`.

## 1   Introduction

Self-supervised methods like DINOv2 [31, 7, 16, 9] have demonstrated remarkable success by leveraging unlabeled data to learn visual representations able to solve many tasks in zero or few shot manners [3]. However, the performances of those methods heavily rely on the availability of clean and high-quality datasets. As a result, some prior works have focused on building data curation pipelines for self-supervised learning (SSL) [37, 1]. But data curation disregards samples and introduces additional design questions and thus begs the following question:

*Can we enable off-the-shelf SSL models like DINOv2 to learn from highly corrupted data?*

In real-world scenarios, datasets often contain noise that can severely degrade the performance of learned representations, and clean references required for supervised denoising are typically unavailable due to sensor limitations, privacy constraints, or acquisition costs.

This is especially common in medical imaging [28, 43, 27], astrophysics [30], geological rock imaging [6], remote sensing [17] and finance [29]. For example, the earth radar images taken by ESA Sentinel-1 [14] comes with speckle noise, and the clean data for supervised denoising is difficult to acquire since it requires better sensor and atmospheric conditions. As another example, Sloan Digital Sky Survey (SDSS) imaging in the COSMOS region contains significant sky noise [30] which can obscure faint celestial objects. Despite advancements in SSL, methods like DINOv2, as shown in Figure 5, still yield significant reduction in performance when trained directly on corrupted datasets.

To address the challenges outlined above, a trivial baseline is a denoiser-preprocessed pipeline, which trains a self-supervised denoiser on the noisy data first, then trains the SSL models on the denoised

39th Conference on Neural Information Processing Systems (NeurIPS 2025).

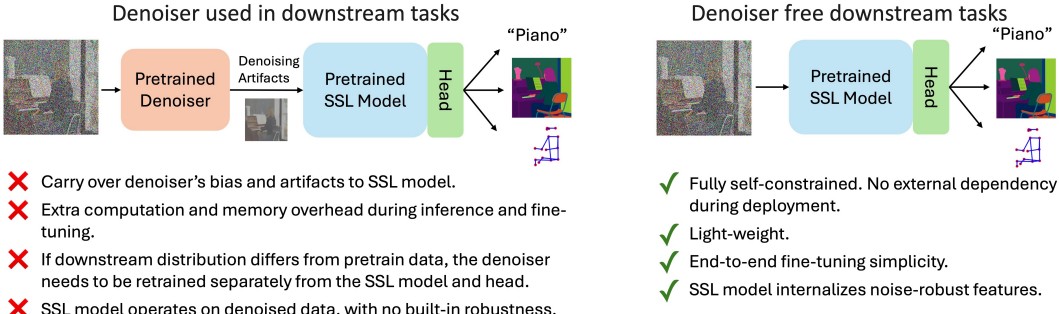

Figure 1: Comparison of downstream pipelines with (left) and without (right) denoisers. The denoiser-free pipeline shows numerous advantages in efficiency, simplicity and robustness.

images. In downstream tasks, the SSL denoiser always serves as a preprocessor because the SSL model is only trained on denoised data thus cannot handle noisy inputs. While effective, this pipeline adds substantial inference latency, fine-tuning overhead, and deployment complexity (Figure 1, left). Instead, we explore training strategies that enable an off-the-shelf SSL backbone (e.g., DINOv2) to ingest noisy inputs directly, thereby ditching the denoiser during downstream use (Figure 1, right). To the best of our knowledge, this problem has little-to-no prior work in the existing literature. We take a first step by corrupting ImageNet [12] with synthetic noise that mirrors real-world degradations, providing a controlled yet challenging testbed for systematic evaluation and benchmarking.

In this paper, we propose a novel, fully self-supervised approach to learn noise-robust representations. Our approach begins similarly as the denoising pipeline baseline: train a SSL denoiser to create a denoised version of the dataset. We then employ a curriculum [5] to train DINOv2 on this denoised dataset for robust feature learning and subsequently restart training on the original noisy dataset to adapt the model to real-world noise. At inference time or downstream task finetuning, the denoiser can be discarded as the SSL model has learned to extract useful representations directly from noisy inputs—without relying on external denoising. In addition, We further introduce a regularization loss using a denoised teacher to guide the student during noisy training, encouraging alignment between noisy and denoised embeddings and improving robustness under extreme noise. Extensive experiments show that our method significantly improves classification and instance recognition performance over DINOv2 trained directly on noisy data. Remarkably, our framework achieves results comparable to, and sometimes exceeding, those obtained by the denoising pipeline baseline. In summary, we make the following contributions:

1. We propose a thorough sensitivity analysis of the latest SOTA SSL method (DINOv2) to noise being present in its pretraining data, a previously unexplored aspect in the literature. We identify that the method is sensitive to its pretraining data quality, especially in low-resource regimes (e.g., short training schedules or smaller datasets).

2. We find that, under mild noise levels, increasing training duration and data size can substantially mitigate performance degradation, but never fully recover the clean-data baseline. This offers actionable guidance for practitioners to deal with noisy data during pretraining.

3. We introduce a new training paradigm for enabling models like DINOv2 to learn robust representations directly from noisy pretraining data. By combining noise curriculum learning with a novel denoised regularization loss, our method achieves fully denoiser-free downstream fine-tuning and inference while maintaining strong performance across tasks. As shown in Figure 2, our methods, DINOv2 w/ NC and DINOv2 w/ NCT improve significantly over the DINOv2 baseline, and closely matches the denoiser-preprocessed pipeline N2N + DINOv2.

## 2   Related Work

**Self-Supervised Learning for Images.**  A large body of work has focused on learning visual representations using self-supervised methods [15, 46, 33, 9, 10]. Among them, joint-embedding methods have emerged as state-of-the-art, training models to produce similar embeddings for different views of the same image. These methods rely on architectural and loss regularizations to prevent representation collapse, such as enforcing cross-correlation constraints [44] or using momentum

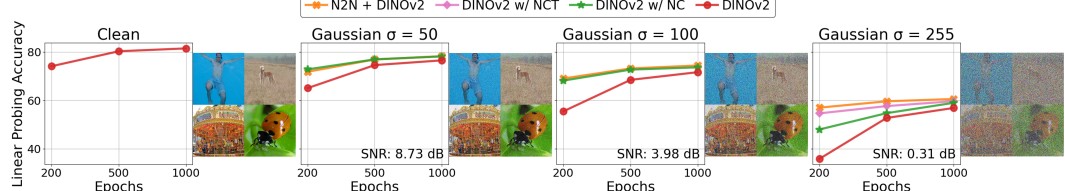

Figure 2: Linear probing accuracies and noise examples on ImageNet-100 dataset. The denoiser-preprocessed baseline N2N + DINOv2 and our denoiser-free methods DINOv2 w/ NC and NCT significantly improve over the DINOv2 baseline. Note that DINOv2 w/ NC and N2N + DINOv2 **overlap** at moderate noise levels, and DINOv2 w/ NCT is only evaluated at Gaussian $\sigma = 255$.

encoders [16, 7, 47, 11]. DINOv2 [31], used in this work, is a joint-embedding method that achieves state-of-the-art performance on ImageNet classification and many evaluation benchmarks.

**Self-Supervised Image Denoising** learns to denoise noisy images without access to noisy-clean pairs. Noise2Noise [26] uses aligned noisy-noisy pairs derived from the same clean image to train a U-Net [35] model. However, obtaining such pairs is not always practical. Neighbor2Neighbor [20], as used in this paper, mitigates such issue by sampling two subimages from a noisy image which serve as a noisy-noisy pair [32]. It achieves competitive performance on a range of denoising benchmarks [45]. Another line of work uses blind-spot networks (BSN) [24, 8] to learn denoise without clean-image supervision. BSN uses the same noisy image as its inputs to supervise its outputs. It either applies masks in inputs [21, 4, 40] or masks in network structure [22, 42] to prevent model collapse.

**Self-Supervised Representation Learning for Noisy Data.** There is very limited work on learning noise-robust features for image data. Most of the past works concern noisy time-series data, such as speech [25, 2, 39, 49] and EEG data [48]. [48] demonstrates applying time series SSL methods on noisy datasets yields poor performance, and proposes denoiser-driven contrastive learning by using a conventional denoiser to create noised-conditioned positive and negative pairs. This approach of leveraging a denoiser to learn noise-robust representation shares similar motivations as our method.

## 3 Noise Robust Self-Supervised Learning

In this section, we introduce our self-supervised method for robust representation learning on noisy image data. We start with a two-stage baseline, then moving towards fully self-supervised, denoiser-free noise robustness through curriculum learning and regularization.

### 3.1 Preliminary: Curriculum Learning Background and Denoiser-Preprocessed Baseline

**Curriculum learning** refers to a training scheme that trains a machine learning model from easier to harder data that imitates the learning order in human curricula. This often leads to faster convergence and better performance. The seminal work [5] formally defines a curriculum as a sequence of training distributions $\langle Q_1, ..., Q_t, ..., Q_T \rangle$. Each distribution $Q_t$ is a reweighting of the target training distribution $P(z)$: $Q_t(z) \propto W_t(z)P(z) \ \forall z : \int Q_t(z)dz = 1$ such that following conditions are satisfied: (1) the entropy of distributions increase over time, i.e., $H(Q_t) < H(Q_{t+1})$. This implies that the model sees more diverse or complex examples as training progresses. A rising entropy indicates reduced certainty and increased difficulty. (2) Unnormalized weighting for any example increases over time, i.e., $W_t(z) \leq W_{t+1}(z) \forall z$. (3) Final distribution equals the target, i.e., $Q_T(z) = P(z)$. In practice, the first condition is most widely preserved, while the latter two are often relaxed to allow more flexible curricula [38]. In this work, we adopt this perspective and design our training schedule to follow an entropy-based curriculum from easy to hard examples.

**N2N + DINOv2: Joint Self-Supervised Denoising and Representation Learning.** As a starting point, we consider a denoiser-preprocessed baseline, an upper bound that combines a self-supervised denoiser with a joint-embedding SSL model. Given a noisy dataset $X$, we first train a denoiser $f_\theta$ on $X$ and generate a denoised set $X_{\text{denoised}} = f_\theta(X)$. This is used to train a self-supervised learner $g_\theta$, such as DINOv2. While effective, this pipeline requires an explicit denoiser in all downstream tasks. We use Neighbor2Neighbor [20] as the denoiser for its robustness across noise levels, but emphasize that the pipeline is **denoiser-agnostic**, a key benefit also shared by our later proposed methods. In practice, the denoiser should be selected based on the domain-specific noise characteristics.

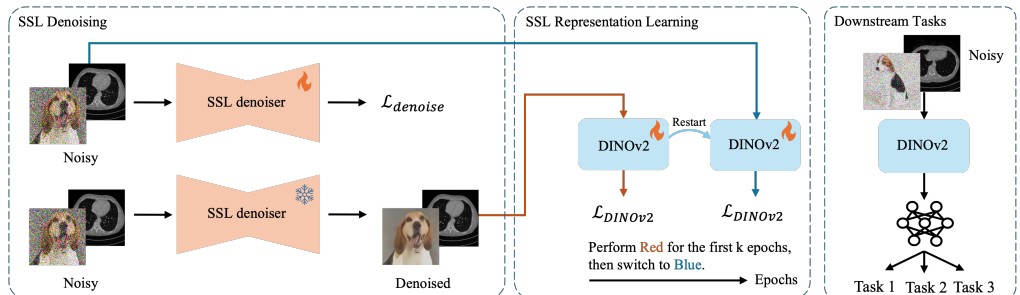

Figure 3: Overview of DINOv2 w/ NC which comprises SSL denoising, SSL representation learning and downstream tasks. A SSL denoiser is trained to denoise the data [28]. Then, DINOv2 is trained on the denoised data for early epochs followed by restarting training on noisy data. Lastly, after some fine-tuning, downstream tasks are performed directly on noisy inputs with an added prediction head.

### 3.2 DINOv2 w/ NC (Noise Curriculum): Curriculum Learning for Self-Supervised Representation Learning

Ideally, instead of relying on a separate denoiser, the representation model should develop robustness to noise on its own, enabling an **end-to-end** training setup in downstream tasks. Figure 3 outlines our curriculum learning approach to achieve this. Following the definition in Section 3.1, we design a curriculum $\langle Q_1, Q_2 \rangle$ that satisfies the key requirement $H(Q_1) < H(Q_2)$. Here, $Q_1$ corresponds to the distribution over *denoised* images $X_{\text{denoised}}$, which lies in a more structured, lower-entropy subspace. $Q_2$ corresponds to the distribution over the original *noisy* images $X$, which has higher entropy due to the injected stochastic corruption that expands the support of the data distribution.

Such a curriculum is further motivated by the following toy example. We consider a simple MLP with 1 hidden layer trained on MNIST [23] using ReLU activation, and optimize the following self-supervised joint embedding loss Equation (1) that applies a soft identity constraint on empirical covariance matrix, which is a canonical formulation that aligns embeddings of different augmented views while preventing representational collapse:

$$\min_{\theta} \quad \frac{1}{n} \sum_{i=1}^{n} \mathbb{E}_{\tau_1, \tau_2 \sim \mathcal{T}} \left\| m_\theta(\tau_1(x_i)) - m_\theta(\tau_2(x_i)) \right\|_2^2 + \lambda \left\| \hat{\Sigma}_\theta - I_k \right\|_F^2 \tag{1}$$

where $\hat{\Sigma}_\theta = \frac{1}{n} \sum_{i=1}^{n} (z_i - \bar{z})(z_i - \bar{z})^\top$, $z_i = m_\theta(\tau(x_i))$, $m$ is the MLP and $\tau$ is the augmentation. Under Gaussian noise ($\sigma = 0.4 \cdot 255$), we observed that: Training and linear probing on the clean train set (50 epochs each) yields $91.21 \pm 0.36\%$ (1-sigma) accuracy on the clean test set. Training and linear probing on the noisy train set (50 epochs each) yields $64.55 \pm 3.47\%$ on the noisy test set. Surprisingly, curriculum learning (training 30 epochs on clean, then 20 on noisy), followed by 50 epochs of linear probing on noisy, recovers performance to $83.05 \pm 0.45\%$ on the noisy test set. This shows that the model can adapt to the noisy data without forgetting what it learned from the clean phase. The significant recovery in performance (see Appendix Figure 6), even with a simple MLP and vector data, suggests that curriculum learning encourages the model to internalize noise-robust features. We now build on this intuition to scale our method to complex vision models like DINOv2:

**Denoised Pretraining**: We start with an initialized self-supervised model $g_\theta$ (DINOv2) for representation learning. We train $g_\theta$ on the denoised set $X_{\text{denoised}}$ for $k$ epochs to establish a stable initialization that captures the underlying structure of the data while reducing the impact of noise.
**Noise Curriculum Transition**: Using the weights learned in the first $k$ epochs, we restart training on the original noisy set $X$ until convergence. Here 'restart' means re-initializing all training dynamics such as learning rate and weight decay scheduling. This enables the model to better adapt to the noisy dataset, learning noise-robust representations from the stable features learned on the denoised data.
**Denoiser-free Downstream Tasks and Inference**: For downstream tasks involving noisy images, such as classification, the representations learned by $g_\theta$ can be directly utilized with a prediction head $h_\theta$ for task-specific learning. Further fine-tuning can be performed directly on the noisy data $z$ without requiring a denoising module, since the training strategy internalizes noise-robust features into the learned representations. This enables the entire model, both backbone $g_\theta$ and head $h_\theta$, to be optimized end-to-end on the downstream objective and deployed without any preprocessing at inference time as shown in Equation (2):

$$\text{Finetuning: } \mathcal{L} = \mathbb{E}_{(z,y)}\left[L(h_\theta(g_\theta(z)), y)\right] \quad \text{Inference: } \hat{y} = h_\theta(g_\theta(z)) \tag{2}$$

This highlights a key advantage of the proposed **DINOv2 w/ NC** as it enables fully denoiser-free downstream fine-tuning and inference. In contrast, the two-stage baseline **N2N + DINOv2** requires a separate self-supervised denoiser trained and applied not only during pretraining but also downstream task fine-tuning and inference. This external dependency adds computational overhead and pipeline complexity, risks transferring the denoiser's bias to the SSL model, and deviates from our goal of learning noise-robust representations directly from corrupted data.

### 3.3 DINOv2 w/ NCT (Noise Curriculum Teacher): Anchored Curriculum Learning for Improved Self-Supervised Representation Learning

Under high or extreme noise levels, a good initialization from denoised training may not be sufficient, because strong noise may destabilize and disturb the prior learned representations during the noisy training. Thus, we propose to utilize the embeddings of the denoised images as anchors to regularize the training on noisy data. Let the teacher backbone of DINOv2 be $T$, and the student backbone be $S$. The original DINOv2 loss can be expressed as

$$L_{\text{dinov2}} = L_{\text{dino\&ibot}}\left(T(\tau_t(x)), S(\tau_s(x))\right) + L_{\text{koleo}} \tag{3}$$

Where $L_{\text{koleo}}$ is the Koleo Regularization, $\tau_t$ is the augmentation for teacher input, and $\tau_s$ is the augmentation for student input. $L_{\text{dino\&ibot}}$ is the combined DINO and iBOT loss that minimizes the cross-entropy of image-level and patch-level output scores between teacher $\{p_t^{\text{img}}, p_t^{\text{patch}}\}$ and student $\{p_s^{\text{img}}, p_s^{\text{patch}}\}$. Here the backbones $T$ and $S$ include 'heads' that output vector scores.

$$T(\tau_t(x)) = \{p_t^{\text{img}}, p_t^{\text{patch}}\} \quad S(\tau_s(x)) = \{p_s^{\text{img}}, p_s^{\text{patch}}\} \tag{4}$$

$$L_{\text{dino\&ibot}} = -\sum p_t^{\text{img}} \log p_s^{\text{img}} - \sum p_t^{\text{patch}} \log p_s^{\text{patch}} \tag{5}$$

Before restarting the noisy training, we first extract the weights of the teacher backbone $T_{\text{dn}}$ that was trained on denoised data and freeze it. As shown in Figure 4, the architecture of the model now contains three components: teacher $T$, student $S$ and frozen denoised teacher $T_{\text{dn}}$. When restarting the training, given a noisy image $x$ and its denoised version $x_{\text{dn}} = f_\theta(x)$, we apply exactly identical augmentations (i.e., same crops, blur, flip...) to $x$ and $x_{\text{dn}}$, so their embeddings are aligned. In addition to the original DINOv2 loss, we encourage the output scores of the student to be close to that of the denoised teacher. This prevents the training from deviating significantly from its cleaner initialization. So the denoised-regularized loss is given as:

$$L = L_{\text{dinov2}} + \lambda L_{\text{dino\&ibot}}\left(T_{\text{dn}}\left(\tau_t\left(x_{\text{dn}}\right)\right), S(\tau_s(x))\right) \tag{6}$$

Where $\lambda$ is a parameter that controls the strength of the regularization. It is worth noting that the weights of $T$ and $T_{\text{dn}}$ are identical at the start of the training. Also, the technique presented is not restricted to only denoised settings. When clean images are available, such method can be used to improve the overall noise-robustness of DINOv2 by simply replacing 'denoised' components with 'clean' components.

It is important to emphasize that this regularization technique is specifically designed to complement our curriculum training pipeline. For the regularization to be effective, the frozen teacher $T_{\text{dn}}$ and the trainable teacher $T$ must have alignment in their output embeddings at the beginning of the restart training. This is accomplished by assigning them identical weights. Using a different frozen teacher, even if trained on the same denoised dataset, would not provide meaningful regularization due to misalignments, as shown in Appendix Figure 9. Like NC, NCT is also denoiser-free at downstream fine-tuning and inference.

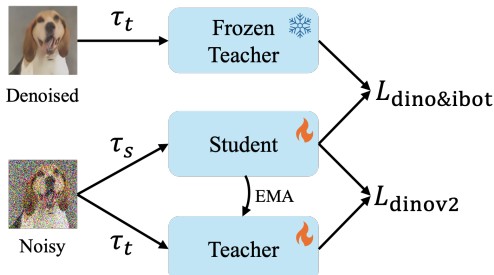

Figure 4: Structure of the denoised regularization loss. In addition to the original DINOv2 loss $L_{\text{dinov2}}$, a regularization loss term $L_{\text{dino\&ibot}}$ is introduced by comparing the output scores between the student and the frozen teacher. The frozen teacher takes in denoised inputs that undergo identical augmentations $\tau_t$ as the noisy inputs of the trainable teacher.

# 4 Experiments

## 4.1 Practical and Challenging Experimental Settings

**Datasets.** We conduct experiments on both ImageNet-100 and ImageNet-1k to evaluate the robustness of SSL methods under noise. ImageNet-100 comprises 100 classes as defined in Mini-ImageNet [36]. The training set includes the first 500 images per class, totaling 50k training and 5k validation images. For large-scale evaluation, we experiment on ImageNet-1k [12] using the full training and validation splits. For instance recognition evaluation, we use the Oxford and Paris dataset [34].

**Noise Addition.** We follow ImageNet-C [19] to introduce three types of noise: Gaussian, Shot (Poisson), and Speckle noise, as these are among the most commonly encountered noise types in natural imaging. For the ImageNet-100 experiments, we use Gaussian noise ($\sigma = 50, 100, 255$; SNR: 8.73 dB, 3.98 dB, 0.31 dB), Shot noise ($\lambda = 10, 3, 1$; SNR: 8.49 dB, 4.11 dB, 0.52 dB), and Speckle noise ($\sigma = 102, 178.5, 255$; SNR: 9.15 dB, 5.58 dB, 3.98 dB). For the ImageNet-1k experiments, we tested Gaussian noise at $\sigma = 100$ and $\sigma = 255$, with corresponding SNRs of 4.36 dB and 0.72 dB respectively. These SNRs are common values reported in real noisy medical imaging [41]. The example visualizations for each noise level are provided in Section D.2. The noises are added to raw images before preprocessing (e.g., resizing) to better reflect practical scenarios such as device or environmental noise. Consequently, while the initial noise level is fixed, the preprocessing steps introduce variations in actual noise levels that challenge the model's ability to adapt effectively. Notably, we generate the noise dataset once prior to training and do not apply noise on-the-fly during data loading, which closely simulate real-world scenarios where one starts with a fixed noisy dataset.

**Implementation Details.** In ImageNet-100 experiments, we use ViT-S/16 as the base architecture for DINOv2, and unless otherwise mentioned, train it for 200 epochs with a batch size of 40 using the AdamW optimizer. The base learning rate is $7.9 \times 10^{-4}$ (scaled by the square root of batch size). The denoiser is trained for 100 epochs with a batch size of 4 using Adam. The linear probe uses a batch size of 128 and is trained for 12.5k steps. In ImageNet-1k, we adopt ViT-B/16 as the backbone and train DINOv2 for 100 epochs with a batch size of 512 and drop path rate of 0.1. The learning rate is $2.8 \times 10^{-3}$. The denoiser is trained on a 100k subset with a batch size of 8 for 50 epochs (for Gaussian $\sigma = 255$) and 90 epochs (for Gaussian $\sigma = 100$). The linear probe uses batch size 1024 for 25k steps. Each training is done on a single RTX 4090 (~8h) or 4×L40S GPUs (~65h). The strength parameters are determined via light parameter sweeps with the details provided in Section A.2.

## 4.2 Linear Probing Reveals Superior Noise Adaptation

We begin by evaluating how different methods adapt to various levels and types of noise using a 200-epoch training schedule on ImageNet-100. We train a linear classifier over a frozen DINOv2 backbone on the training set, and report classification performance over the validation set. We compare the classification performance of the DINOv2 backbone over four training strategies. (1) **N2N + DINOv2**: train the backbone on denoised images for 200 epochs and evaluate the model on denoised images (2) **DINOv2 w/ NC**: train the backbone on denoised images for 140 epochs, then restart training on noisy images for 60 epochs without regularization. Finally, evaluate the model on noisy images. The rationale for such partitions of epochs is provided in Appendix B.3. (3) **DINOv2 w/ NCT**: same as the DINOv2 w/ NC but with the regularization loss applied in the 60 epochs of noisy training. We only evaluate the NCT

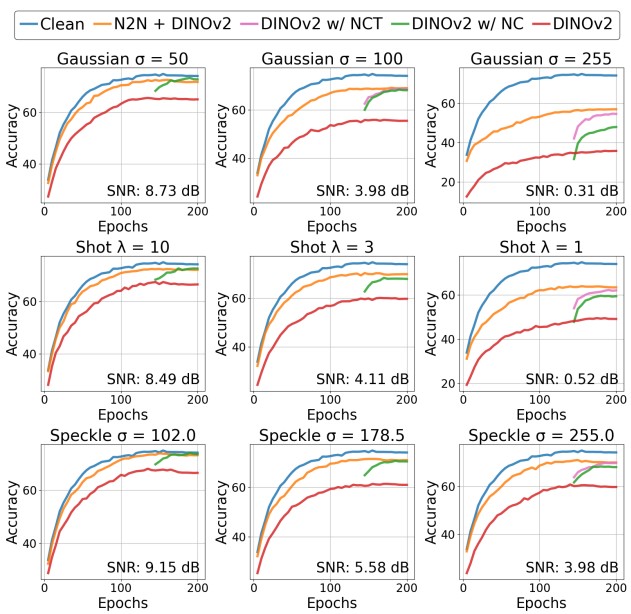

Figure 5: Linear probing classification accuracies on DINOv2 backbone weights saved every 5 epochs during training on ImageNet-100 under various noise levels.

Table 1: Performance comparison of different methods when scaling training duration and data size.

| Noise Type | Method | 200 Ep. | 500 Ep. | 1000 Ep. |
|---|---|---|---|---|
| Clean | DINOv2 | 74.1 | 80.3 | 81.4 |
| Gaussian-50 SNR: 8.73 dB | N2N + DINOv2 | 71.8 | 76.9 | 78.2 |
| | DINOv2 w/ NC | 72.8 | 76.9 | 78.1 |
| | DINOv2 | 65.0 | 74.5 | 76.5 |
| Gaussian-100 SNR: 3.98 dB | N2N + DINOv2 | 69.0 | 73.2 | 74.4 |
| | DINOv2 w/ NC | 68.1 | 72.7 | 73.6 |
| | DINOv2 | 55.4 | 68.4 | 71.6 |
| Gaussian-255 SNR: 0.31 dB | N2N + DINOv2 | 57.0† | 59.6‡ | 60.5 |
| | DINOv2 w/ NCT | 54.6 | 57.6† | 59.6‡ |
| | DINOv2 w/ NC | 47.9 | 54.7 | 59.0 |
| | DINOv2 | 35.8 | 52.7 | 56.8 |

(a) Evolution of linear probing accuracies on ImageNet-100 under extended training durations. **Relative ranking among methods remain consistent across epochs**. See visualization in Figure 2.

| Method | Clean | Gaussian-100 SNR: 4.36 dB | Gaussian-255 SNR: 0.72 dB |
|---|---|---|---|
| N2N + DINOv2 | - | 73.1 | 57.2 |
| DINOv2 w/ NCT | - | 72.1 | 55.8 |
| DINOv2 w/ NC | - | 70.9 | 53.5 |
| DINOv2 | 79.0 | 70.7 | 51.0 |
| SimCLR | 60.1 | 48.0 | 34.2 |

(b) Linear probing classification accuracy on ImageNet-1k under Gaussian noise using a 100-epoch training schedule. **DINOv2 w/ NCT** and **DINOv2 w/ NC** show substantial improvements over the **DINOv2** baseline, particularly at high noise levels ($\sigma = 255$). SimCLR is included here as a reference baseline.

method on the four most noisy scenarios (lowest SNRs), i.e., Gaussian noise ($\sigma$=255, $\sigma$=100), Shot noise ($\lambda$=1), and Speckle noise ($\sigma$=255). (4) **DINOv2**: train the backbone on noisy images for 200 epochs, and evaluate the model on noisy images. Additionally, we train and evaluate the model on original clean images, referred to as **Clean**. Figure 5 illustrates the classification accuracy curves.

As shown in Figure 5, our proposed method (green) significantly outperforms the backbone trained solely on noisy images (red) across all noise types and noise levels. At moderate noise levels, i.e., the first two columns, the accuracy of our method closely matches and even exceeds the backbone that is trained and evaluated on explicitly denoised data (shown in orange lines). Only at extreme noise levels, i.e., the last column, the gap between **DINOv2 w/ NC** and **N2N + DINOv2** widens, primarily due to the significant loss of the original image signal. For instance, as shown in Section D.2, with Gaussian noise at $\sigma = 255$, many images become virtually unrecognizable to the naked eye. Hence, regularization is introduced to guide the model in these extremely noisy scenarios. As illustrated in Figure 5, regularization **DINOv2 w/ NCT** improves the model's performance at all four scenarios. The improvement amplifies as the gap between **N2N + DINOv2** and **DINOv2 w/ NC** widens, with a substantial improvement of 6.7% over **NC** achieved at Gaussian noise with $\sigma = 255$. However, when **N2N + DINOv2** is on par with **DINOv2 w/ NC**, the regularization provides diminishing benefit, as demonstrated with Gaussian noise at $\sigma = 100$. This is expected since the guidance becomes redundant when its representation quality is comparable to or lower than that of the original model.

## 4.3 Scaling Improves Performance Across Datasets

Next, we scale both training and data size to evaluate performance in more resource-rich settings. **Longer Training Duration.** We further improve the performance of our methods by extending the training on ImageNet-100 to longer epochs. For 500-epoch configuration, we restart the training from the 200-epoch denoised checkpoint (i.e., N2N + DINOv2) as described in Section 4.2; for the 1000-epoch configuration, we restart the training from the 500-epoch denoised checkpoint. As shown in Table 1a and Figure 2, accuracies increase as epoch increases and the relative ranking among methods remains consistent across training durations: **N2N + DINOv2** achieves the highest accuracy, followed closely by **DINOv2 w/ NC or NCT**, while the baseline **DINOv2** remains the lowest. This demonstrates the scalability of the proposed methods. Interestingly, we observe that DINOv2 w/ NCT consistently converges to or outperforms its anchor's accuracy, as indicated by † and ‡ in Table 1a, demonstrating that the regularization objective successfully guides the model toward robust representations. Notably, the performance gap between DINOv2 and the other methods narrows as training progresses, suggesting that DINOv2 is sensitive to noise and benefits significantly from longer training. In contrast, our noise curriculum and regularization approach greatly accelerates convergence, achieving similar or better performance in roughly half the training time.

**Larger Dataset.** In our controlled ImageNet-1k experiment, we constrain the DINOv2 backbone to undergo a total of 100 training epochs to enable fair comparison across methods. **DINOv2 w/ NC**: train the backbone on denoised images for 30 epochs followed by 70 epochs of training on noisy images. **DINOv2 w/ NCT**: While we find that restarting at 30 epochs yields the best performance for DINOv2 w/ NC, the denoised teacher at this early stage lacks sufficiently strong representations to serve as an effective regularizer. To address this, we continue training the denoised teacher to 100 epochs, then freeze it for regularizing the DINOv2 backbone for 70 epochs of noisy training. Since

Table 2: Comparison of linear probing accuracies on clean and noisy validation sets for models trained on noisy ImageNet-1k and ImageNet-100 datasets. **Surprisingly, our DINOv2 w/ NC or NCT method outperforms the two-stage N2N + DINOv2 on clean validation sets,** demonstrating more accurate and generalizable representations.

| Dataset | Training Noise | Method | Clean | Noisy |
|---|---|---|---|---|
| ImageNet-1k (100 epochs) | Gaussian-100 SNR: 4.36 dB | N2N + DINOv2 | **75.8** | 73.1 |
| | | DINOv2 w/ NCT | 75.2 | 72.1 |
| | | DINOv2 w/ NC | 74.3 | 70.9 |
| | | DINOv2 | 74.1 | 70.7 |
| | Gaussian-255 SNR: 0.72 dB | N2N + DINOv2 | 64.1 | 57.2 |
| | | DINOv2 w/ NCT | **65.6** | 55.8 |
| | | DINOv2 w/ NC | 63.7 | 53.5 |
| | | DINOv2 | 61.6 | 51.0 |
| ImageNet-100 (1000 epochs) | Gaussian-50 SNR: 8.73 dB | N2N + DINOv2 | 78.5 | 78.2 |
| | | DINOv2 w/ NC | **79.4** | 78.1 |
| | | DINOv2 | 78.8 | 76.5 |
| | Gaussian-100 SNR: 3.98 dB | N2N + DINOv2 | 76.0 | 74.4 |
| | | DINOv2 w/ NC | **76.7** | 73.6 |
| | | DINOv2 | 74.6 | 71.6 |
| | Gaussian-255 SNR: 0.31 dB | N2N + DINOv2 | 56.7 | 60.5 |
| | | DINOv2 w/ NCT | **59.4** | 59.6 |
| | | DINOv2 w/ NC | 57.9 | 59.0 |
| | | DINOv2 | 58.8 | 56.8 |

Table 3: Linear evaluation performance of applying noise curriculum (NC) to other SSL models on ImageNet-100. Performance is improved for all tested models, **highlighting the broader applicability of our approach.** The accuracy curve over epochs are visualized in Figure 12 in Section B.5.

| SSL Model | Architecture | Method | Accuracy |
|---|---|---|---|
| SimCLR | ResNet50 | N2N + SimCLR | 64.3 |
| | | SimCLR w/ NC | 61.1 |
| | | SimCLR | 59.0 |
| MoCo v3 | ViT-S | N2N + MoCo v3 | 60.4 |
| | | MoCo v3 w/ NC | 55.3 |
| | | MoCo v3 | 52.2 |
| SimSiam | ResNet50 | N2N + SimSiam | 68.4 |
| | | SimSiam w/ NC | 65.7 |
| | | SimSiam | 64.8 |
| iBOT | ViT-S | N2N + iBOT | 61.9 |
| | | iBOT w/ NC | 62.7 |
| | | iBOT | 56.9 |
| DINO | ViT-S | N2N + DINO | 62.5 |
| | | DINO w/ NC | 62.1 |
| | | DINO | 57.9 |
| DINOv2 | ViT-S | N2N + DINOv2 | 69.0 |
| | | DINOv2 w/ NC | 68.1 |
| | | DINOv2 | 55.4 |

the frozen teacher and the DINOv2 backbone are derived from the same training run and pass through the same 30-epoch state, alignment is preserved. This allows us to meet the 100-epoch constraint while still leveraging a stronger teacher. Table 1b shows that **DINOv2 w/ NCT** improves over the **DINOv2** baseline with substantial absolute gains of $1.4\%$ and $4.8\%$ at noise levels $\sigma = 100$ and $\sigma = 255$, respectively. **DINOv2 w/ NC** also achieves a strong improvement of $2.5\%$ under the more challenging $\sigma = 255$ setting. The relatively smaller gain of DINOv2 w/ NC at $\sigma = 100$ is due to the fact that, in large-scale settings with moderate noise, longer training alone significantly improves DINOv2's robustness. Since DINOv2 w/ NC allocates a portion of the training budget to denoised pretraining, it shortens the noisy training phase, slightly limiting its benefit. Overall, while scaling training on large datasets improves DINOv2's noise adaptation under moderate noise, denoised pretraining and regularization remain essential under high noise to maintain strong performance. **As an additional baseline**, we also evaluate DINOv2 trained exclusively on clean images and directly tested on noisy inputs. The results presented in Section C.1 confirm that clean-pretrained model suffers severe accuracy drop, which underscores the importance of our noise-aware training.

### 4.4 Probing on Clean Test Set Highlights Better Representation Quality

In previous sections, linear classification accuracies were measured on validation sets that had the same noise distribution as the training set. To further assess representation quality, we evaluate performance on the clean, original ImageNet validation set. Note that N2N + DINOv2 is evaluated without the denoiser because the clean images are already noise-free. Performance on this clean validation set serves as a strong indicator of how well a model captures representations of the "noise-free" real world. **Surprisingly, as highlighted by bold numbers in Table 2, DINOv2 w/ NC or NCT outperforms N2N + DINOv2 in most cases**, strongly suggesting that our denoiser-free approach learns more robust and generalizable representations than the two-stage denoising pipeline. This can be attributed to the fact that explicit denoising in the two-stage approach inevitably loses some useful information, leading to a degradation in representation quality in the second stage. This finding has important real-world implications: in scenarios where large-scale noisy data is readily available for pretraining, but only a limited amount of clean data is available for specific downstream tasks, DINOv2 w/ NC and NCT offer more effective solutions for learning transferable representations.

### 4.5 Instance Recognition Validates Versatility

To further validate the versatility of our method, we extend the evaluation to another downstream task. Here, we examine the instance-level recognition task on noisy images using the embeddings output by the backbone. The approach is non-parametric as the images' embeddings are ranked based on their

Table 4: Mean Average Precision (mAP) on instance-level recognition tasks using Oxford and Paris datasets. **DINOv2 w/ NC** consistently outperforms **DINOv2** and closely matches **N2N + DINOv2** across benchmarks, with notable gains in extreme noise scenarios using regularization. **This validates the broader applicability of our approach to tasks beyond classification.**

| Noise | Method | Oxford | Paris | Noise | Method | Oxford | Paris | Noise | Method | Oxford | Paris |
|---|---|---|---|---|---|---|---|---|---|---|---|
| Gaussian $\sigma$=50 | N2N + DINOv2 | 20.89 | 38.62 | Shot $\lambda$=10 | N2N + DINOv2 | 21.36 | 40.00 | Speckle $\sigma$=102 | N2N + DINOv2 | 22.01 | 40.47 |
| | DINOv2 w/ NC | **21.30** | **40.94** | | DINOv2 w/ NC | **21.94** | **42.20** | | DINOv2 w/ NC | **22.40** | **41.28** |
| | DINOv2 | 19.15 | 40.04 | | DINOv2 | 21.13 | 39.73 | | DINOv2 | 20.33 | 39.04 |
| Gaussian $\sigma$=100 | N2N + DINOv2 | **21.83** | 39.04 | Shot $\lambda$=3 | N2N + DINOv2 | 21.40 | **40.39** | Speckle $\sigma$=178.5 | N2N + DINOv2 | **21.31** | **39.31** |
| | DINOv2 w/ NC | 20.13 | **40.14** | | DINOv2 w/ NC | **22.11** | 39.12 | | DINOv2 w/ NC | 21.21 | 38.74 |
| | DINOv2 | 16.39 | 32.54 | | DINOv2 | 18.59 | 35.65 | | DINOv2 | 18.63 | 37.06 |
| Gaussian $\sigma$=255 | N2N + DINOv2 | 15.67 | 33.68 | Shot $\lambda$=1 | N2N + DINOv2 | 18.96 | **37.22** | Speckle $\sigma$=255 | N2N + DINOv2 | 20.11 | 39.19 |
| | DINOv2 w/ NCT | **17.07** | **33.73** | | DINOv2 w/ NCT | **20.66** | 36.57 | | DINOv2 w/ NCT | 20.81 | **39.31** |
| | DINOv2 w/ NC | 14.81 | 29.97 | | DINOv2 w/ NC | 19.25 | 36.67 | | DINOv2 w/ NC | **21.10** | 38.59 |
| | DINOv2 | 7.37 | 18.51 | | DINOv2 | 16.37 | 32.60 | | DINOv2 | 20.11 | 35.63 |
| **(a) Gaussian Noise** | | | | **(b) Shot Noise** | | | | **(c) Speckle Noise** | | | |

cosine similarity with a query image's embedding. The results are based on the medium difficulty evaluation setup in the Oxford and Paris datasets, where both easy and hard images are treated as positive images. We measure mean average precision (mAP) and report results in Table 4. We see that the results closely align with Section 4.2. **DINOv2 w/ NC** greatly outperforms **DINOv2** in all comparisons. The mAP of **DINOv2 w/ NC** is on par with **N2N + DINOv2**, even at extreme noise levels. For example, at Shot $\lambda = 1$ and Speckle $\sigma = 255$, the mAP of **DINOv2 w/ NC** surpasses that of **N2N + DINOv2** on the Oxford dataset. **DINOv2 w/ NCT** outperforms all other methods the majority of the time in high noise settings, again validating its effectiveness.

### 4.6  DINOv2 w/ NC is Applicable to Diverse SSL Models

To test the applicability of DINOv2 w/ NC beyond DINOv2, we extend the DINOv2 w/ NC to various other SSL models on ImageNet-100, including SimCLR [9], MoCo v3 [11], SimSiam [10], iBOT [47], and DINO [7]. These models encompass both contrastive and non-contrastive approaches, as well as ViT-based [13] and CNN-based architectures [18]. All experiments are conducted with Gaussian ($\sigma$=100) noise without denoised regularization. As shown in Table 3, DINOv2 w/ NC improves over the noisy baseline across all models. The improvement is modest or minor for the first three models, i.e., SimCLR, MoCo v3, and SimSiam. The variation in contrastivity and backbone architecture among these models suggests that neither factor is critical to the method's effectiveness. In contrast, iBOT and DINO (both are precursors to DINOv2) show substantial gains, similar to the improvements observed in DINOv2, where the **SSL Model w/ NC** closely matches **N2N + SSL Model**. This indicates that DINOv2 w/ NC has broad applicability, with more pronounced benefits for models sharing architectural and loss-function similarities with DINOv2. We also benchmark the performance of these models on the instance recognition task in Table 8 of Section B.5, which shows exactly identical trends as the linear evaluation, further reaffirming the generality of our approach. Detailed training configurations are provided in the Section A.4.

## 5  Limitations, Conclusion and Future Work

In this work, we proposed a fully self-supervised framework for learning noise-robust visual representations. Leveraging an SSL denoiser and the curriculum training strategy enables DINOv2 to improve significantly upon the noisy baseline. While our benchmark focuses on synthetic noise due to the scarcity of large-scale *labeled noisy* datasets that allows for standard downstream evaluation, this choice enables controlled, reproducible assessment across noise types. One limitation is the assumption that self-supervised denoiser can denoise reasonably well; while it is a fair assumption as we show in Section B.6 that even a very weak denoiser (1 training epoch) can still substantially improve accuracy, it remains an open dependency that could affect performance in more challenging settings. Another key direction for future research is to develop automated, adaptive strategies for curriculum schedule, as the current design requires tuning to determine when to shift from denoised to noisy samples during pretraining. More broadly, our methodology can be extended to other modalities like time series data, including noisy audio, EEG signals and financial data; we aim to establish a unified SSL approach for robust representation learning.

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

**Outline of the Appendix:**

# A    Technical Details about Experiments

## A.1    Details of Toy Experiment in Section 3.2

In both pretraining and linear probing evaluation, we use Adam optimizer, batch size 256, and a constant learning rate of $1.0 \times 10^{-4}$. The augmentation includes random resized crop and random affine, as shown in Listing 1. The strength $\lambda$ for soft identity constraint is set to 0.5.

The results over 17 runs are visualized in Figure 6. We observe that adopting a noise curriculum not only significantly improves accuracy, **but also substantially stabilizes performance**, as evidenced by the narrower green box compared to the wider orange box. This highlights the dual benefits of curriculum learning in enhancing both robustness and consistency.

```
transforms.Compose([
    transforms.RandomResizedCrop(28, scale=(0.8, 1.0)),
    transforms.RandomAffine(
        degrees=15,
        translate=(0.1, 0.1),
        scale=(0.95, 1.05),
        shear=10
    ),
])
```
Listing 1: Data augmentation used in pretraining for the simple MLP with 1 hidden-layer in the toy experiment.

## A.2    Regularization Strength Details and Ablation

For results in Figure 5, we select optimal regularization strength based on light sweeps visualized in Figure 7. Specifically, $\lambda$ is set to 1.1, 5.0, 2.0 and 4.0 for Gaussian $\sigma = 255$, Shot $\lambda = 1$, Speckle $\sigma = 255$ and Gaussian $\sigma = 100$ respectively. Notably, Figure 7 shows that increasing the strength

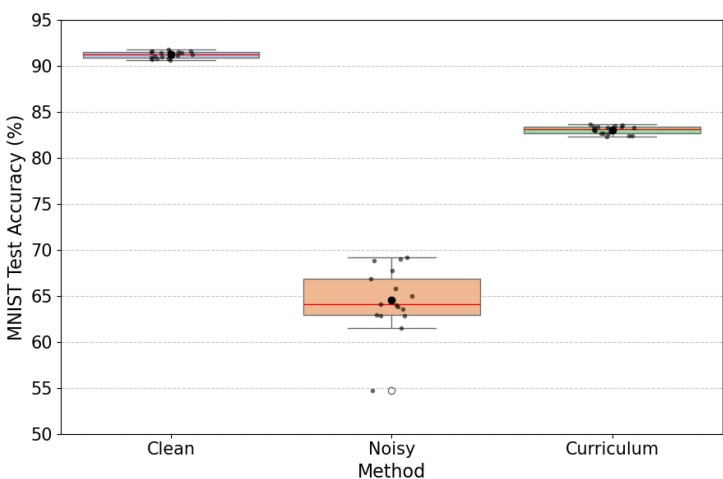

Figure 6: Box plots of toy experiments accuracies over 17 runs. "Clean" for MLP trained and tested on clean. "Noisy" for MLP trained and tested on noisy. "Curriculum" for MLP w/ NC tested on noisy. **Using noise curriculum not only significantly improves accuracy, but also stablizes performance, as shown by the narrower green box compared to the wider orange box.**

results in a sharp increase in accuracy followed by fluctuatons around a higher plateau, demonstrating the effectiveness of NCT.

In scaling epoch experiment for ImageNet-100 (Table 1a), $\lambda$ is set to 1.1, 0.5 and 0.2 for 200, 500 and 1000 epochs respectively. We find longer training duration requires smaller regularization strength. This agrees with our main finding that scaling training can partially mitigate the performance degradation, thus requires less guidance from denoised teacher. In ImageNet-1k experiment (Table 1b), $\lambda$ is set to 2.0 and 1.6 for Gaussian $\sigma = 100$ and Gaussian $\sigma = 255$.

For evaluation on clean set (Table 2), the strengths are set to the same value as their noisy counterparts detailed in the preceding paragraph.

In instance recognition experiments (Table 4), $\lambda = 0.2$ for all settings as we find it generally yields better results. For mixed noise-clean data evaluation shown later in Section B.4, $\lambda$ is set to 1.1.

### A.3   Training Details for Main DINOv2 Experiments

**Training Details for Section 4.2.** For curriculum-based training, although the model is trained on denoised images for only 140 epochs, the training dynamics in the first stage, such as learning rate scheduling, are configured based on the total training duration of 200 epochs. In the second stage of training on noisy images, the training dynamics are configured for a total duration of 60 epochs to ensure smooth convergence.

**Training Details for Section 4.3.** In ImageNet-1k experiments (Table 1b), the training configuration for **DINOv2** and **N2N + DINOv2** are set as default, consistent with its original implementation. In the noisy-stage training of **DINOv2 w/ NC** and **DINOv2 w/ NCT**, we reduce the linear learning rate warm-up epochs from 10 to 5, and the teacher temperature warm-up epochs from 30 to 15 to accommodate the reduced training duration for noisy adaptation under the constraint of 100 epochs. Intuitively, less warm-up is needed because denoised pre-training provides a good initialization.

**Training Details for SimCLR Baseline in Table 1b.** Consistent with DINOv2, we set batch size to 512 and train for 100 epochs during pre-training. We use the open-source PyTorch implementation from AndrewAtanov/simclr-pytorch. Since SimCLR typically requires more epochs during linear evaluation compared to DINOv2, we train the linear head for 90 epochs (112.59k steps) with a batch size of 1024 to ensure fair comparison.

**Details of Linear Probing.** Consistent with the original DINOv2 paper, when performing linear probing, a light grid search is conducted over hyperparameters, i.e., learning rate, output layer, and

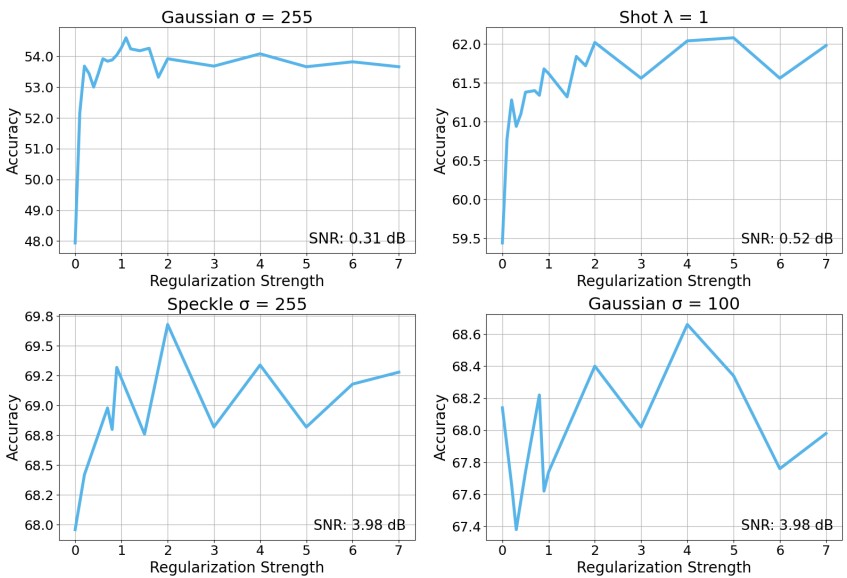

Figure 7: Regularization strength ablation to determine optimal strength for DINOv2 w/ NCT in Figure 5. Most plots show a **rapid increase in linear probing accuracy followed by fluctuations around a higher plateau as strength increases.** Due to computational constraints, the sweep was not performed with uniform resolution across all settings and may vary across noise types.

whether concatenating average-pooled tokens. The highest accuracy value is reported which is a common practice.

## A.4 Training Details for Other SSL Models in Section 4.6

The linear evaluations of all models train the linear head for 50 epochs.

**SimCLR.** During training, we use a batch size of 100 and LARS optimizer. After square root scaling, the base learning rate is set to $0.075 \times \sqrt{100} = 0.75$. During linear evaluation, we use a batch size of 200 and SGD optimizer. After square root scaling, the base learning rate is set to $1.6 \times \sqrt{200/4096} = 0.35$.

**MoCo v3.** During training, we use a batch size of 128 and AdamW optimizer. After square root scaling, the base learning rate is set to $4 \times 10^{-4} \times \sqrt{128/1024} = 1.414 \times 10^{-4}$. During linear evaluation, we use a batch size of 200 and SGD optimizer. After linear scaling, the base learning rate is set to $0.2 * (200/256) = 0.15625$.

**SimSiam.** During training, we use a batch size of 100 and SGD optimizer. After square root scaling, the base learning rate is set to $0.15 \times \sqrt{100/256} = 0.09375$. During linear evaluation, we use a batch size of 200 and LARS optimizer. We found the learning rate provided in SimSiam's codebase is too small for effectively training the linear head. After some experimentation, we set the base learning rate to 2.34375 after the linear scaling $3.0 \times (200/256)$.

**iBOT.** During training, we use a batch size of 80 and AdamW optimizer. After square root scaling, the base learning rate is set to $5 \times 10^{-4} \times \sqrt{80/256} = 2.795 \times 10^{-4}$. During linear evaluation, we use a batch size of 128. We also adopt the sweeping hyperparameter implementation in iBOT to obtain the best linear evaluation value. This involves sweeping over a range of learning rates, and a set of candidate optimizers including LARS and SGD.

**DINO.** During training, we use a batch size of 128 and AdamW optimizer. After square root scaling, the base learning rate is set to $5 \times 10^{-4} \times \sqrt{128/256} = 3.54 \times 10^{-4}$. During linear evaluation, we use a batch size of 128 and SGD optimizer. After square root scaling, the base learning rate is set to $0.001 \times \sqrt{128/256} = 7.1 \times 10^{-4}$.

## A.5 Noise Addition Formula

As stated in Section 4.1, we follow ImageNet-C [19] to introduce Gaussian, Shot, and Speckle noises to the images. Let original image be $x$, and normalized image be $\tilde{x} = x/255$.

Gaussian noise:
$$x_{\text{noisy}} = 255 \cdot \text{clip}\left(\tilde{x} + \mathcal{N}\left(0, c^2\right), 0, 1\right) \tag{7}$$
where $c = \frac{\sigma}{255}$, and $\sigma \in \{50, 100, 255\}$.

Shot (Poisson) noise:
$$x_{\text{noisy}} = 255 \cdot \text{clip}\left(\frac{\text{Poisson}(\tilde{x} \cdot \lambda)}{\lambda}, 0, 1\right) \tag{8}$$
where $\lambda \in \{10, 3, 1\}$.

Speckle noise:
$$x_{\text{noisy}} = 255 \cdot \text{clip}\left(\tilde{x} + \tilde{x} \cdot \mathcal{N}(0, c^2), 0, 1\right) \tag{9}$$
where $c = \frac{\sigma}{255}$, and $\sigma \in \{102, 178.5, 255\}$.

Note that the clipping is applied to the $[0, 1]$ range in all noise types, which further increases the difficulty of denoising at high noise levels. A significant portion of the noise signal may get clipped and hence irreversibly distorted or lost.

## A.6 Signal-to-Noise Ratio Details

All SNR values given are calculated from the raw noisy images without preprocessing, e.g., resizing, crop. We report the average SNR between the training set and the validation set since they have no significant difference. In the instance recognition, we report the average between the Oxford and Paris datasets. Detailed per-set SNR values are provided in Table 5 and Table 6.

Table 5: SNR values of training and validation sets for ImageNet-100 and ImageNet-1k datasets.

| Dataset | Noise Type | Train SNR (dB) | Val SNR (dB) |
|---|---|---|---|
| ImageNet-100 | Gaussian $\sigma = 50$ | 8.84 | 8.63 |
| | Gaussian $\sigma = 100$ | 4.09 | 3.87 |
| | Gaussian $\sigma = 255$ | 0.43 | 0.19 |
| | Shot $\lambda = 10$ | 8.55 | 8.44 |
| | Shot $\lambda = 3$ | 4.16 | 4.05 |
| | Shot $\lambda = 1$ | 0.58 | 0.47 |
| | Speckle $\sigma = 102$ | 9.17 | 9.14 |
| | Speckle $\sigma = 178.5$ | 5.59 | 5.56 |
| | Speckle $\sigma = 255$ | 4.00 | 3.96 |
| ImageNet-1k | Gaussian $\sigma = 100$ | 4.41 | 4.31 |
| | Gaussian $\sigma = 255$ | 0.78 | 0.67 |

Table 6: SNR values for Oxford and Paris datasets.

| Noise Type | Oxford SNR (dB) | Paris SNR (dB) |
|---|---|---|
| Gaussian $\sigma = 50$ | 8.86 | 9.04 |
| Gaussian $\sigma = 100$ | 4.10 | 4.27 |
| Gaussian $\sigma = 255$ | 0.39 | 0.55 |
| Shot $\lambda = 10$ | 8.66 | 8.79 |
| Shot $\lambda = 3$ | 4.24 | 4.39 |
| Shot $\lambda = 1$ | 0.65 | 0.78 |
| Speckle $\sigma = 255$ | 4.09 | 4.17 |
| Speckle $\sigma = 178.5$ | 5.69 | 5.75 |
| Speckle $\sigma = 102$ | 9.28 | 9.31 |

# B  Ablation Studies

## B.1  Ablation Unveils Synergy in Curriculum Learning

We conduct ablation studies on ImageNet-100 to analyze the contribution of the two key components of the curriculum learning method, specifically in the context of linear probing classification on images with Gaussian noise ($\sigma = 100$).

**Only Denoised Pretraining.** We train DINOv2 on denoised images for 140 epochs, then **continues** to train it on noisy images for 60 epochs without resetting training dynamics.

**Only Noisy Training with Restart.** We train DINOv2 on noisy images for 140 epochs, then **restart** training on the same noisy set for 60 epochs.

Figure 8 illustrates that denoised pretraining contributes nearly twice the accuracy improvement over the noisy baseline compared to restarting training. Notably, their effects are not merely additive, as our combined method outperforms the sum of their individual gains, highlighting a synergistic interaction between the two techniques. The synergy arises because, while denoised pretraining provides robust initial weights, the diminished learning rate in later stages limits adaptation to noisy images. Restarting training resets the learning rate, allowing the model to fully adapt to noise and leverage the learned representations effectively.

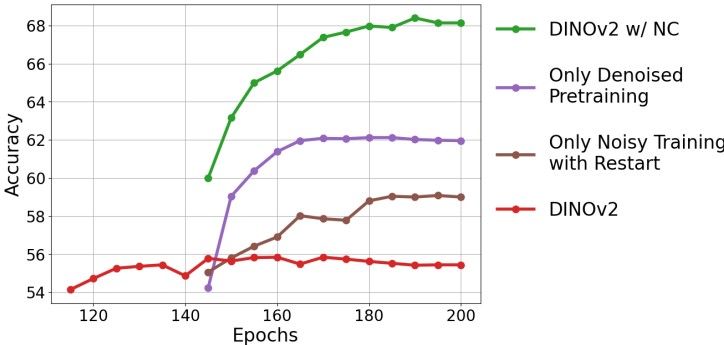

Figure 8: Comparison of curriculum learning ablations showing the contributions of denoised pretraining and restarting noisy training to linear evaluation accuracy. **"Only Denoised Pretraining" (purple) provides nearly double the improvement of "Only Noisy Training with Restart" (brown) over the Noisy baseline (red).**

## B.2  Critical Roles of Alignment and Initialization

To evaluate the design choices of our regularization loss, we conduct ablation studies on ImageNet-100 under extreme Gaussian noise ($\sigma = 255$), where the regularization loss has the most significant impact. These studies provide insights into the importance of alignment and initialization in our approach.

**Unaligned Frozen Teacher.** Instead of extracting the weights of the current teacher backbone at the end of the denoised training phase, we retrain a separate backbone on the same denoised dataset to extract teacher weights. This alternative frozen teacher is then used during the noisy training phase. Unlike the original method, this introduces misaligned embeddings between the frozen teacher, the trainable teacher, and the student. At regularization strength 0.2, as shown by the brown line in Figure 9, this setup results in a marginal performance improvement of just 1.1% over the green baseline, significantly lower than the aligned case depicted by the purple line. At regularization strength 1.1, as shown in Figure 10, the strong unaligned teacher instead destabilizes training, dragging accuracy downward as learning proceeds, ultimately underperforming even the DINOv2 w/ NC. These results highlight the seamless integration between the regularization loss and the curriculum training pipeline, where the first stage's outputs are essential for effectively regularizing the second stage.

**Randomly Initialized Backbone.** To further investigate the role of denoised pretraining, we replaced the trainable teacher and student backbones with randomly initialized weights during the noisy training phase while keeping the frozen teacher unchanged. At regularization strength 0.2, this configuration yields extremely poor performance, with accuracy dropping below 20%, as shown

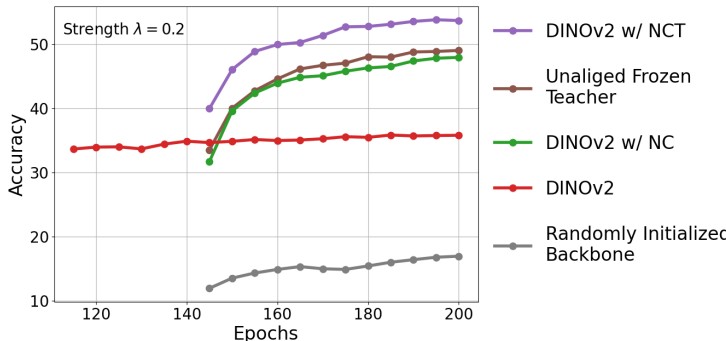

Figure 9: Comparison of regularization ablations at strength = 0.2 showing the contribution of alignment and initialization to linear evaluation accuracy. Aligned regularization (purple) significantly outperforms unaligned regularization (brown) and random initialization (grey). **This illustrates both the alignment and the denoised pretraining contributes substantially to the effectiveness of the regularization.**

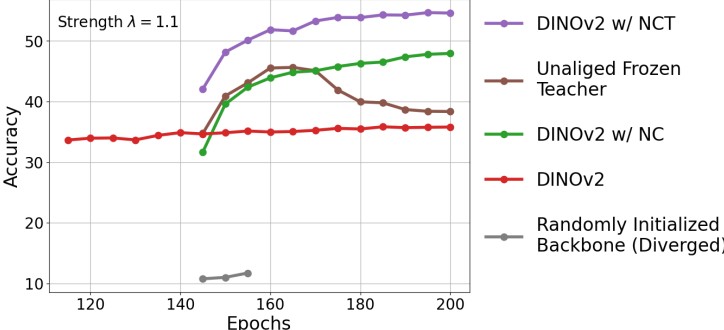

Figure 10: Comparison of regularization ablations at strength = 1.1 showing the contribution of alignment and initialization to linear evaluation accuracy. Aligned regularization (purple) significantly outperforms unaligned regularization (brown) and random initialization (grey). **Random initialization diverges after 30 epochs of training. Strong unaligned regularization drags the accuracy down as training progresses.**

by the gray line in Figure 9. Notably, at regularization strength 1.1, the training diverges after 30 epochs as the loss explodes, as shown by the discontinuous grey line in Figure 10. These results highlight the necessity of denoised pretraining, even in the presence of a regularization term, as it provides a crucial foundation for the training process. Without it, the training fails to leverage the frozen teacher's guidance.

### B.3 Restart at Different Epochs

We investigated the impact of restarting training on noisy data at different stages: (1) training on denoised data for 140 epochs, followed by 60 epochs on noisy data (a common setting in this paper); (2) 130 epochs on denoised data, then 70 epochs on noisy data; and (3) 120 epochs on denoised data, then 80 epochs on noisy data. As shown in Figure 11, the performances across these variations are similar, with no approach consistently outperforming the others. Thus, we decide to use 140 epochs as the restart point.

### B.4 Enduring Effectiveness in Mixed Noisy-Clean Data

In real-world scenarios, datasets often contain images of varying quality, posing challenges for SSL methods. In this section, we investigate how different noise-clean distributions on ImageNet-100 affect the SSL model's performance and the effectiveness of our method under these distributions. We replace $x\%$ (i.e., $0\%, 2\%, 10\%$) of noisy data with clean images. So in our noise curriculum, the model is trained on a mix of $(100 - x)\%$ denoised and $x\%$ clean images in the first stage, then on a mix of $(100 - x)\%$ noisy and $x\%$ clean images in the second stage. The validation set is also adjusted to follow the same distribution as the training set. From Table 7, we can see that generally

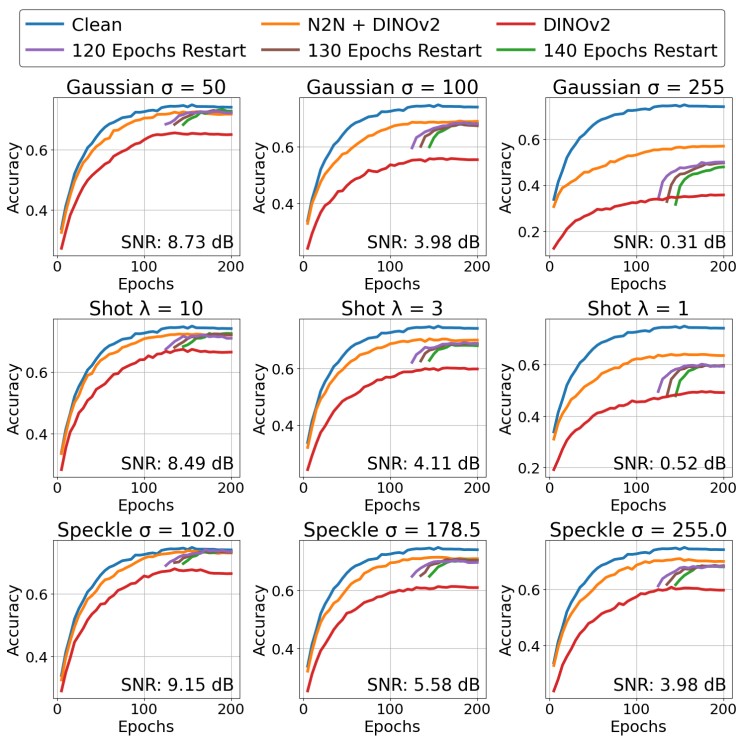

Figure 11: Linear Probing Classification accuracy for different restarting epochs. **We observe that no approach consistently outperforms the others between restarts at 120, 130 and 140 epochs.**

the linear evaluation accuracy decreases across all methods when more clean data is introduced but noisy data remains the majority, which can be attributed to a reduced overall representation alignment between noisy, denoised and actual clean data. On the other hand, through column-wise comparisons, **DINOv2 w/ NC** consistently outperforms **DINOv2** and closely matches **N2N + DINOv2**; regularized method **DINOv2 w/ NCT** brings substantial improvement over unregularized method. It is clear that our curriculum learning framework and denoised regularization are still effective despite the varying training distributions.

Table 7: Mixed Noise-Clean Data Evaluation: Impact of introducing varying percentages of clean data (0%, 2%, 10%) on linear evaluation accuracy. We observe that increasing the proportion of clean data generally reduces accuracy across all methods due to reduced representation alignment. **However, our DINOv2 w/ NC and NCT are still effective despite the varying training distributions.**

| Noise Type | Method | 0% Clean | 2% Clean | 10% Clean |
|---|---|---|---|---|
| | N2N + DINOv2 | 71.8 | 72.0 | 71.1 |
| Gaussian-50 | DINOv2 w/ NC | 72.8 | 72.3 | 71.6 |
| | DINOv2 | 65.0 | 61.2 | 61.9 |
| | N2N + DINOv2 | 69.0 | 67.8 | 67.7 |
| Gaussian-100 | DINOv2 w/ NC | 68.1 | 66.6 | 67.0 |
| | DINOv2 | 55.4 | 54.8 | 53.1 |
| | N2N + DINOv2 | 57.0 | 55.2 | 53.5 |
| Gaussian-255 | DINOv2 w/ NCT | 54.6 | 53.8 | 51.7 |
| | DINOv2 w/ NC | 47.9 | 47.3 | 47.3 |
| | DINOv2 | 35.8 | 36.9 | 35.8 |

Table 8: Mean Average Precision (mAP) on instance-level recognition tasks for other SSL models. The pattern is consistent with Table 3. Noise curriculum (NC) improves performance for all models.

| SSL Model | Method | Oxford - M | Paris - M |
|---|---|---|---|
| | N2N + SimCLR | 21.43 | 35.32 |
| SimCLR | SimCLR w/ NC | 19.08 | 33.39 |
| | SimCLR | 18.04 | 31.65 |
| | N2N + MoCo v3 | 15.34 | 29.52 |
| MoCo v3 | MoCo v3 w/ NC | 13.79 | 30.05 |
| | MoCo v3 | 12.95 | 27.72 |
| | N2N + SimSiam | 24.03 | 40.10 |
| SimSiam | SimSiam w/ NC | 21.12 | 39.77 |
| | SimSiam | 20.64 | 38.15 |
| | N2N + iBOT | 12.26 | 25.52 |
| iBOT | iBOT w/ NC | 14.60 | 29.79 |
| | iBOT | 11.05 | 24.05 |
| | N2N + DINO | 15.04 | 29.16 |
| DINO | DINO w/ NC | 16.01 | 31.33 |
| | DINO | 12.73 | 27.70 |
| | N2N + DINOv2 | 21.83 | 39.04 |
| DINOv2 | DINOv2 w/ NC | 20.13 | 40.14 |
| | DINOv2 | 16.39 | 32.54 |

## B.5 Further Results for Other SSL Models

Figure 12 shows the accuracy over epochs for SSL models tested in Section 4.6. Table 8 shows the performance of the SSL models on the instance recognition task using the Oxford and Paris dataset. Both results illustrate that the noise curriculum improves the performance of all SSL models on noisy data, demonstrating the wide applicability of the method.

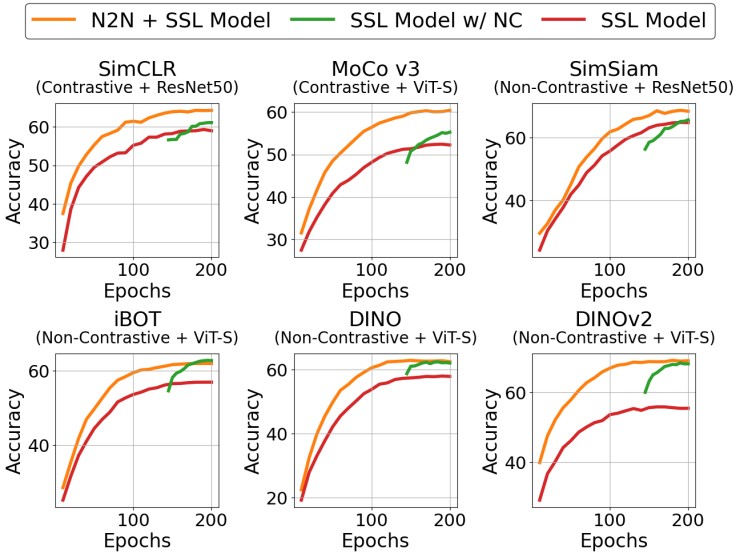

Figure 12: Linear evaluation performance over epochs of applying DINOv2 w/ NC to other SSL models on ImageNet-100. Noise curriculum (green) improves performance for all tested models, with substantial gains observed in iBOT and DINO variants, showing similar trend as DINOv2.

## B.6 Robustness to Denoiser Quality

To investigate the level of tolerance of DINOv2 w/ NC to the denoiser quality, we conduct ablation studies on how the number of denoiser training epochs impacts the SSL model's final performance. We use the number of training epochs as a quantifiable proxy for denoiser effectiveness. The evaluation follows our default setting: 200 epochs of DINOv2 ViT-S training with Gaussian-100 noisy data. The results are shown in Table 9. We observe that even training the N2N denoiser for just 1 epoch leads to a substantial improvement of DINOv2 w/ NC over the DINOv2 baseline (i.e., 10.7 increase), while being only 2 percent less than that of a well-trained denoiser (100 epochs). Although the 1-epoch denoiser still produces many undesirable artifacts like a very weak denoiser, the curriculum still yields substantial improvements in downstream performance. This highlights the robustness of our curriculum, which has considerable tolerance to the denoiser's quality. A denoiser has to be unreasonably bad (e.g., training diverged or collapsed) to provide no return in performance.

Table 9: Performance of DINOv2 w/ NC on ImageNet-100 with the N2N denoiser trained for varying epochs. Notably, even a weak denoiser trained for only 1 epoch substantially improves DINOv2 w/ NC over DINOv2.

| Method | Denoiser Training Epochs | Accuracy |
|---|---|---|
| DINOv2 | / | 55.4 |
| DINOv2 w/ NC | 1 epoch | 66.1 |
| DINOv2 w/ NC | 5 epochs | 67.4 |
| DINOv2 w/ NC | 100 epochs | 68.1 |

Table 10: Performance of clean-pretrained DINOv2 models under different noise conditions. Linear probes are fine-tuned either on clean or noisy data while keeping the backbone frozen. **The clean-pretrained DINOv2 suffers severe drop in performance when applied on the noisy test set.**

| Training Dataset | Architecture | Eval Dataset | Eval Noise | Probe Tuned on Clean | Probe Tuned on Noisy |
|---|---|---|---|---|---|
| LVD-142M | ViT-B/14 | ImageNet-1k | Gaussian-100 | 63.2 | 70.3 |
| | | | Gaussian-255 | 8.9 | 29.0 |
| | | | Clean | **84.5** | / |
| LVD-142M | ViT-S/14 | ImageNet-1k | Gaussian-100 | 49.2 | 61.0 |
| | | | Gaussian-255 | 4.9 | 23.1 |
| | | | Clean | **81.1** | / |
| ImageNet-1k (100 epochs) | ViT-B/16 | ImageNet-1k | Gaussian-100 | 40.7 | 58.1 |
| | | | Gaussian-255 | 2.2 | 20.5 |
| | | | Clean | **79.0** | / |
| ImageNet-100 (1000 epochs) | ViT-S/16 | ImageNet-100 | Gaussian-50 | 60.0 | 74.3 |
| | | | Gaussian-100 | 22.3 | 61.3 |
| | | | Gaussian-255 | 3.1 | 34.5 |
| | | | Clean | **81.4** | / |

Table 11: Linear probing accuracy on noisy ImageNet-100 test set when training on mixed noisy+denoised data. Mixing denoised images results in a 7.4 point drop compared to training on noisy data alone.

| Training Setting | Accuracy |
|---|---|
| 100% Gaussian-100 | 55.4 |
| 50% Gaussian-100 + 50% Denoised | 48.0 |

## C   Baseline Comparisons

### C.1   Performance of Clean-Pretrained Model on Noisy Data

We evaluate the noise-robustness of a DINOv2 model trained exclusively on clean images, in order to reinforce the necessity of noisy training. Two strategies are employed to cover complementary baselines. (1) With the backbone frozen, we fine-tune a linear probe on noisy data drawn from the same distribution as the test set, and then evaluate the full model on noisy data. (2) With the backbone again frozen, we fine-tune a linear probe on clean data and subsequently evaluate the model on noisy data. The first strategy simulates a common downstream scenario where a frozen feature extractor is adapted to a new domain via task-specific heads, while the second serves as a more standard adversarial benchmark that tests the model's robustness to noise without ever exposing it to noise during training. In addition to our own trained models, we also benchmark the officially released DINOv2 weights, which were pretrained on the LVD-142M dataset (142 million images). The results in Table 10 show substantial performance drops (15–70 points) when applying clean-pretrained DINOv2 to noisy test sets. The degradation is particularly severe on Gaussian-255 noise, where the accuracy of ViT-B/14 drops from 84.5 to 29.0 (a 55.5-point decrease), even when the probe is tuned on Gaussian-255 noisy data.

Notably, despite the smaller model size and significantly reduced training data, our DINOv2 w/ NCT (Table 1b) surpasses the official ViT-B/14 weights trained on LVD-142M at both Gaussian-100 (72.1 vs. 70.3) and Gaussian-255 (55.8 vs. 29.0). These results emphasize that standard DINOv2 is not robust to input noise, underscoring the necessity of incorporating noisy images during pretraining to achieve strong downstream performance in noisy settings.

### C.2   Baseline Performance on Mixed Noisy–denoised Data

Another intuitive baseline is to investigate whether mixing denoised samples with noisy ones during training can improve representation learning by increasing the diversity and expressiveness of inputs. Table 11 shows the linear probing accuracies of DINOv2 ViT-S on the noisy ImageNet-100 test set,

when trained for 200 epochs on 100% noisy data (from Table 1(a)) versus on 50% noisy + 50% denoised data. We observe a significant drop of 7.4 points in accuracy when noisy and denoised data are randomly mixed during training. As a result, mixing clean (denoised) and noisy images during training leads to degraded performance, primarily due to reduced representation alignment between noisy and denoised samples, which hinders the model's ability to learn consistent features. This shows naive mixing is suboptimal, and reinforces the necessity of using a staged curriculum learning approach that separates noisy and denoised images.

# D   Visualizations

## D.1   Training Loss Visualization

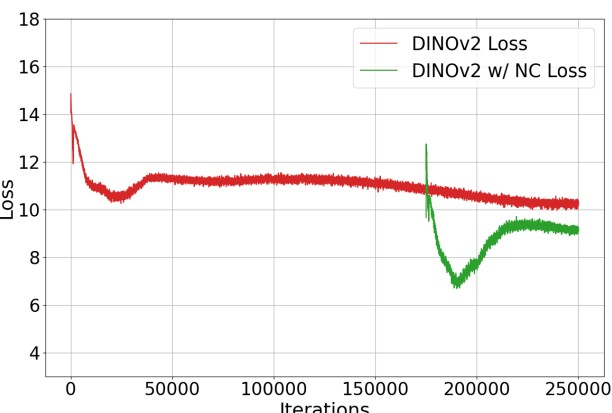

Figure 13: Comparison of training loss between DINOv2 and DINOv2 w/ NC. **DINOv2 w/ NC can descend to a lower loss due to its better initialization from the denoised pretraining.**

Figure 13 shows the comparison between training loss of DINOv2 and DINOv2 w/ NC on the ImageNet-100 noisy data with Gaussian noise ($\sigma = 100$). DINOv2 w/ NC loss (green) descends to a lower value than DINOv2 (red). The two curves are similar in shape which reflects the restarting training technique in our noise curriculum learning.

## D.2   Noisy Images Visualization

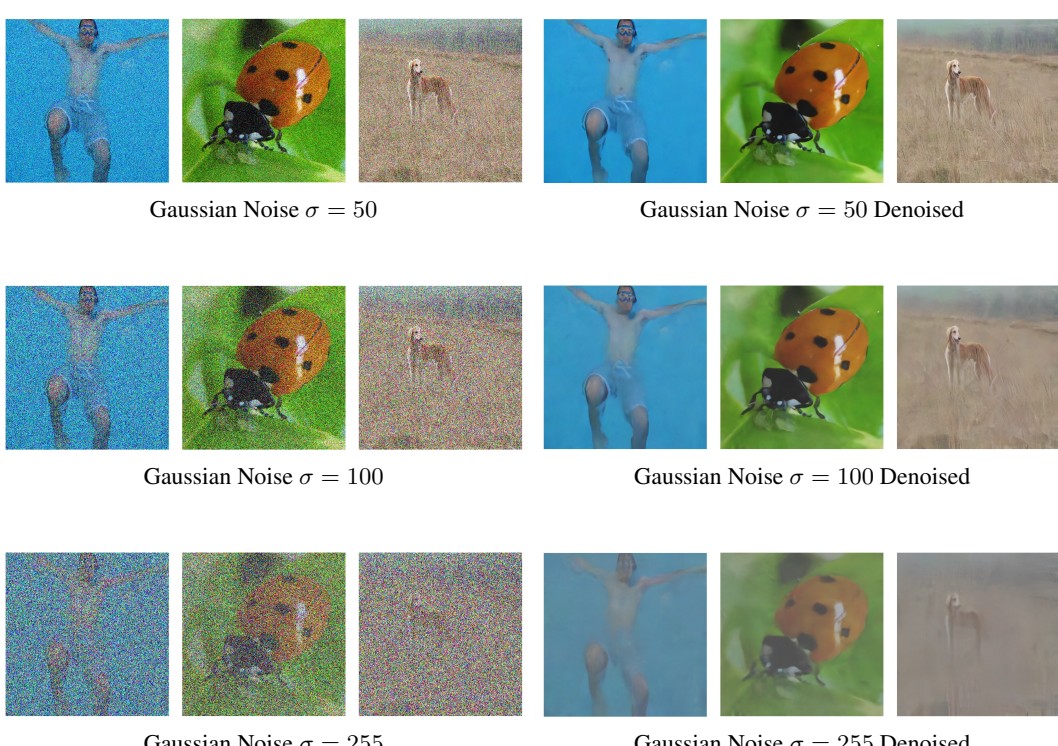

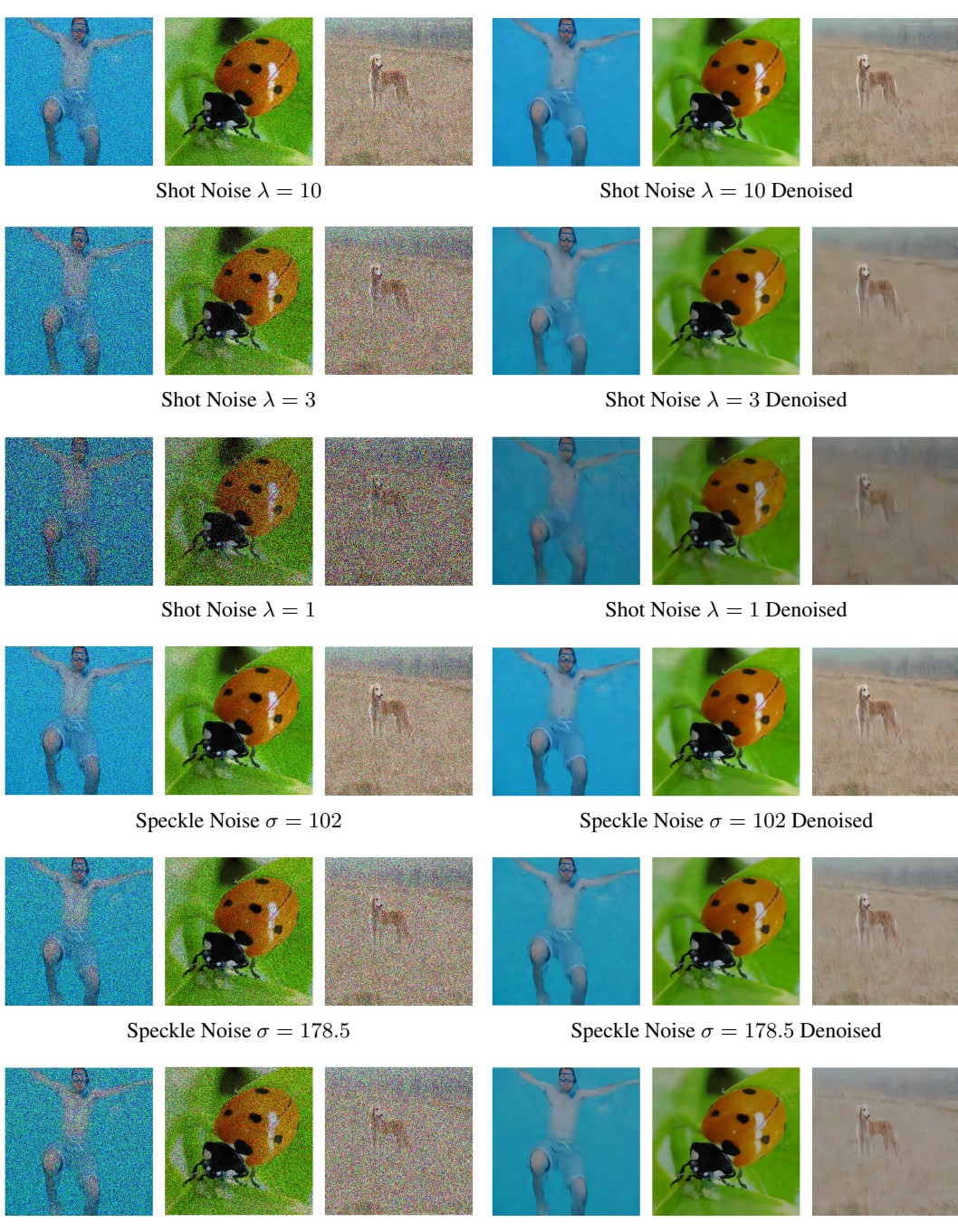

Shot Noise $\lambda = 10$ | Shot Noise $\lambda = 10$ Denoised

Shot Noise $\lambda = 3$ | Shot Noise $\lambda = 3$ Denoised

Shot Noise $\lambda = 1$ | Shot Noise $\lambda = 1$ Denoised

Speckle Noise $\sigma = 102$ | Speckle Noise $\sigma = 102$ Denoised

Speckle Noise $\sigma = 178.5$ | Speckle Noise $\sigma = 178.5$ Denoised

Speckle Noise $\sigma = 255$ | Speckle Noise $\sigma = 255$ Denoised

## D.3 PCA Visualization

To visualize the dense features, we apply PCA to the feature space and map the first three principal components to the RGB channels. We use the ViT-S 200-epoch Gaussian-255 checkpoints from Table 1a for visualization. As shown in , the DINOv2 visualizations (middle of each triplet) fail to clearly separate salient objects from the background, and the resulting color patterns lack semantic consistency across images, suggesting weaker representation quality. In contrast, the DINOv2 w/ NCT visualizations (right of each triplet) distinctly separate salient objects, and objects of the same class are consistently represented with similar color patterns across different images, indicating semantically aligned features. These results demonstrate that NCT yields a substantial improvement in feature quality over the DINOv2 baseline.

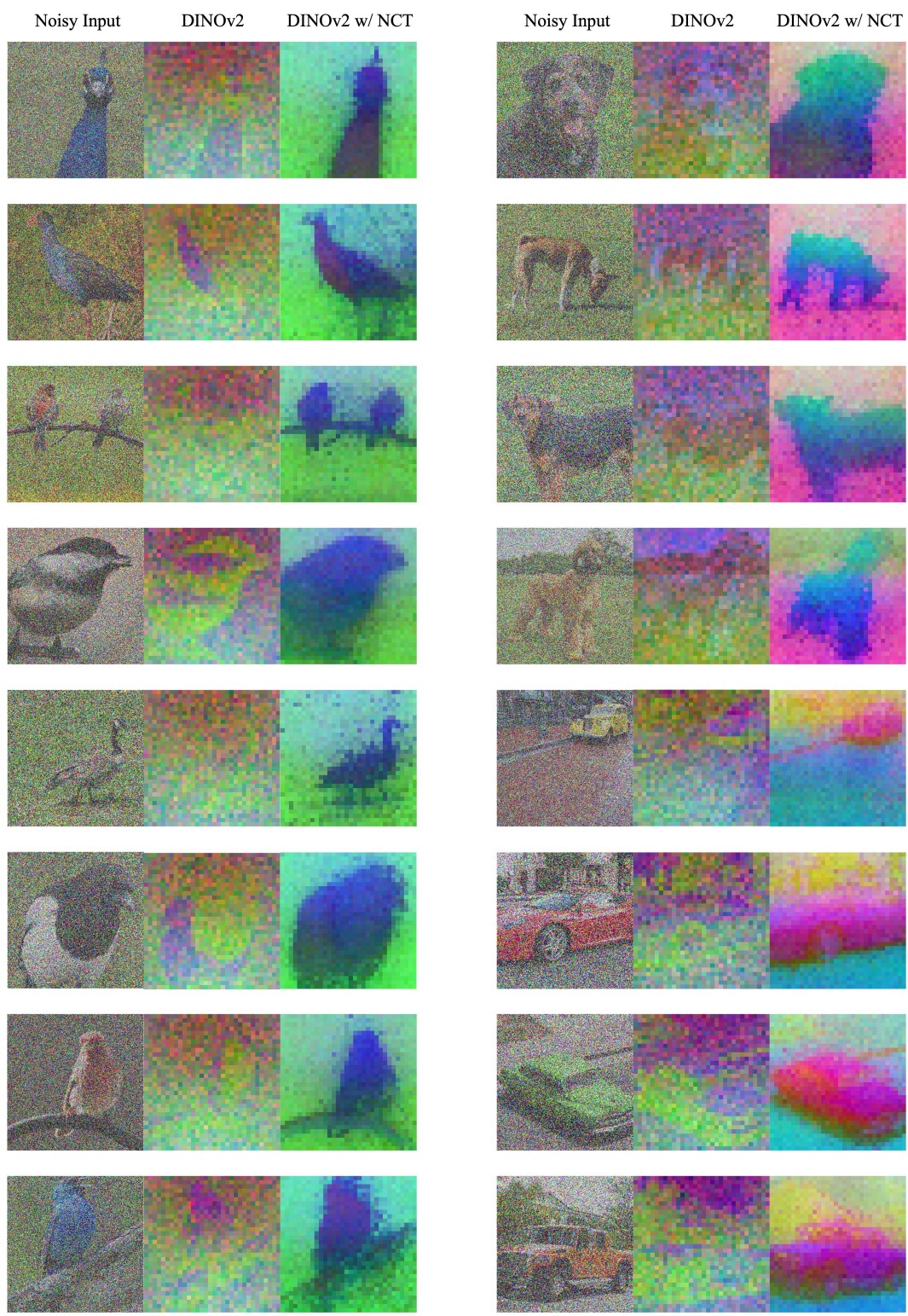

Figure 23: PCA visualizations: we visualize dense features of the models by performing PCA on the feature space and mapping the top three principal components to RGB. In each triplet, the left image is the noisy input with Gaussian noise at $\sigma = 255$. The middle and right images are produced by DINOv2 and DINOv2 w/ NCT (ViT-S, 200 epochs) respectively.

