# OpenReview forum: "Ditch the Denoiser: Emergence of Noise Robustness in Self-Supervised Learning from Data Curriculum"
_NeurIPS.cc/2025/Workshop/Reliable_ML — NeurIPS 2025 - Reliable ML Workshop_

### Official Review · Reviewer_pZya · 2025-09-18
**Relevant and novel paper, hard to understand need for two-stage model however**

**Rating:** 7
**Confidence:** 3

**Review:**

Summary: The paper claims to propose a SSL method for noisy data that directly ingest the noisy data and learns representations, as opposed to needing a denoiser during inference time to denoise the data prior to SSL.

Strengths: This work is very relevant to the conference, and appears to be quite novel based on the authors statement that there is no one who has attempted this specific way of handling noisy data for SSL. There are many comparisons presented that validate and situate this model in the context of SSL methods for noisy data. The results generally show the proposed model performs better than DINOv2 on noisy data and on par with DINOv2 on clean data.

Weaknesses / Limitations: It seems to me that the proposed method is still not better than the combined 2 stage model of the denoiser + DINO v2. The 2 stage architecture seems to do slightly better on noisy data, and as far as I can tell, there are no results showing that the sing stage architecture is better or faster during inference compared to the two stage architecture. Perhaps this could be addressed to strengthen the paper.

Suggestions for Authors: The tables in the results section are a bit hard to parse. It would be helpful for the reader if certain values or rows could be bolded to indicate which one is the best performing model.

---

### Official Review · Reviewer_qcpJ · 2025-09-20
**Good paper**

**Rating:** 7
**Confidence:** 4

**Review:**

Summary:
This paper presents a fully self-supervised framework that enables noise-robust representation learning without requiring a denoiser at inference or downstream fine-tuning. It constructs a denoised-to-noisy data curriculum and combines a teacher-guided regularization, encouraging the model to internalize noise robustness and remove the denoiser.

Strength:
The proposed approach is both intuitive and technically sound. The use of a denoised-to-noisy data curriculum (DINOv2 w/ NC) is a effective strategy, which provides the model with a stable, clean starting point before exposing it to the complexities of noisy data, which helps the model internalize robust features. The denoised teacher regularization (DINOv2 w/ NCT) handles extreme noise levels where a simple curriculum is insufficient. This method anchors the noisy embeddings to their denoised counterparts, effectively guiding the model's learning process.

Experiments are comprehensive and ablation studies are included.

Weaknesses:

Although the framework is denoiser-free at inference, it still relies on a self-supervised denoiser during the initial training phase. The paper acknowledges this as a limitation but the quality of this initial denoiser is a critical dependency.

The experiments are conducted primarily on synthetic noise. It would be beneficial to see results on a more diverse set of real-world noisy datasets from fields like medical imaging or geophysics to fully validate the framework's effectiveness.